# On the Importance of Asymmetry in the Phenotypic Expression of the Genetic Code upon the Molecular Evolution of Proteins

**Marco V. José *** and **Gabriel S. Zamudio**

Theoretical Biology Group, Instituto de Investigaciones Biomédicas, Universidad Nacional Autónoma de México, Ciudad Universitaria 04510, Mexico; gazaso_92@comunidad.unam.mx

*** Correspondence: marcojose@biomedicas.unam.mx

**Abstract:** The standard genetic code (SGC) is a mapping between the 64 possible arrangements of the four RNA nucleotides (C, A, U, G) into triplets or codons, where 61 codons are assigned to a specific amino acid and the other three are stop codons for terminating protein synthesis. Aminoacyl-tRNA synthetases (aaRSs) are responsible for implementing the SGC by specifically amino-acylating only its cognate transfer RNA (tRNA), thereby linking an amino acid with its corresponding anticodon triplets. tRNAs molecules bind each codon with its anticodon. To understand the meaning of symmetrical/asymmetrical properties of the SGC, we designed synthetic genetic codes with known symmetries and with the same degeneracy of the SGC. We determined their impact on the substitution rates for each amino acid under a neutral model of protein evolution. We prove that the phenotypic graphs of the SGC for codons and anticodons for all the possible arrangements of nucleotides are asymmetric and the amino acids do not form orbits. In the symmetrical synthetic codes, the amino acids are grouped according to their codonicity, this is the number of triplets encoding a given amino acid. Both the SGC and symmetrical synthetic codes exhibit a probability of occurrence of the amino acids proportional to their degeneracy. Unlike the SGC, the synthetic codes display a constant probability of occurrence of the amino acid according to their codonicity. The asymmetry of the phenotypic graphs of codons and anticodons of the SGC, has important implications on the evolutionary processes of proteins.

**Keywords:** standard genetic code; symmetry; asymmetry; anticodon code; phenotypic graphs; protein evolution

## 1. Introduction

The decipherment of the standard genetic code (SGC) is a landmark achievement in biological sciences [1,2]. The SGC is a nearly universal map of 61 nucleotide triplets (codons) to 20 amino acids plus two punctuation marks (three stop and one start signals). The SGC became an abstract mathematical problem even before its discovery [3,4]. Symmetrical properties of the SGC have been found [5–8]. Protein synthesis is the outcome of a complex translation system that involves ribozymes, ribosomal proteins, aminoacyl-tRNA synthetases (aaRSs), elongation and termination factors, and three kinds of RNA molecules, to wit, messenger RNA (mRNA), ribosomal RNA (rRNA), and transfer RNA (tRNA) [9,10]. The evolution of tRNAs, aaRSs, and the SGC has been thoroughly examined and reviewed elsewhere [11,12]. Different models for the evolution of the genetic code have been proposed based on different properties of the amino acids and the triplets [13]. Some models consider an RNY primeval code from which the SGC can be derived by frameshift reading mistranslations and/or transitions and transversions in the first or third nucleotide of each codon [7,14,15]. Other models consider

the metabolic pathways for the incorporation of amino acids to the code [16]. Other propositions are the co-evolution model theory [17–20], and Trifonov's consensus model [21].

The 20 encoded amino acids exhibit unique physicochemical properties, which facilitate folding, catalysis, and solubility of proteins, and confer adaptive value to organisms able to encode them [22]. Experimental scientists are generally not attracted/interested in theoretical works, which are rarely cited by biologists as they are mathematically abstract and often divorced from biological context [23]. Despite connections between the mathematical models and data [24], theoretical approaches are still regarded as speculative work [25]. It has been suggested that part of the problem lies in the fact that theorists focused on the table of the SGC [26] and failed to address the central question: co-ordinate evolution of aaRS gene sequences and their cognate tRNAs with the codon assignments [25]. To our knowledge, there are theoretical works that tackle the anticodon and operational codes that are found embedded in tRNA molecules [27–29].

In previous works [7,30], Genetic Hotels of codons and Hotels of amino acids (three-dimensional models) were used to test hypotheses about the evolution of the SGC [14,24]. The usual representation of the SGC as a table allows the visualization of the wobble effect that confers robustness to the genetic code by relating similar codons to the same amino acid, most commonly allowing variation in the third position of the codon triplets and fixing the other two bases, with the exception of the hexa-codonic amino acids. The mathematical representation of this effect has been described previously [15] with the computation of the group of automorphisms of the 6D model of the codons that maintain invariant all the equivalent classes of the codons given by the genetic code. It was shown that these groups are not trivial, thus, providing a theoretical representation of the wobbling effect. In this work, we are concerned with the organization of the amino acids in the SGC.

In the present work, we compare the SGC with the standard tRNA code (S-tRNA-C), which comprises 45 tRNAs (an A in the first anticodon position does not exist and there are no anticodons for stop codons). Codon-anticodon pairing takes place according to wobbling rules in which anticodons recognize more than one codon [26]. Hence, the number of required anticodons is reduced substantially. Overall, anticodons beginning with purines (R) are always of only one kind for one amino acid; this is usually G, sometimes a modified purine. This remarkable wobbling property permits that two or more neighboring codon triplets share a common anticodon. Degeneracy is the known property of SGC of having different numbers of codons specifying each amino acid, also named codonicity. For example, Methionine and Tryptophan are specified by a single codon; and Leucine, Arginine, and Serine are encoded by six triplets. The other 15 amino acids have intermediate values with their codonicity ranging from two to four.

In this work, we set out to search for symmetries of the phenotypic graphs of codons and anticodons, and to discern the biological meaning of symmetry. As the phenotypic graphs (PHGs) of anticodons and codons were found to be asymmetric, we designed synthetic and symmetrical codes (sy$^2$-codes) with the same degeneracy of the SGC.

We applied a neutral model of evolution to proteins [31] as they would be obtained by the biological anticodon code and the built-in sy$^2$-codes. In the sy$^2$-codes, subsets of amino acids formed orbits according to their codonicity, whilst in the natural code, 20 orbits were observed, i.e., one orbit for each amino acid.

**Definition 1.** *The orbit of an element $x \in E$, is defined as $Orb(x) := \{y \in E : \exists\, g \in G : y = g * x\}$, where * denotes the group action. Hence $Orb(x) = G * x$. The latter means that the orbit of an element is all its possible images or destinations under the group action.*

**Definition 2.** *Let $\Re$ be the relation on E defined as $\forall\, x, y \in E : x\Re y \Leftrightarrow \exists\, g \in G : y = g * x$, , where * denotes the group action. The orbit of $x$, denoted by $Orb(x)$, is the equivalence class of $x$ under $\Re$.*

The implication of the asymmetry now becomes apparent: the occurrence of each amino acid in protein evolution is independent of the presence/absence of the remaining 19 amino acids. This means

that the process of molecular evolution applied to proteins sequences becomes free and independent from the strict rules dictated by the SGC due to the selected asymmetry—or lack of symmetry—of the graph of codons and anticodons. In other words, the asymmetry of the anticodon code is disassociated with the deterministic character of the SGC.

## 2. Materials and Methods

The SGC has been modeled upon a 6D hypercube as template using group theory [7,15,32]. The vertices of the hypercube represent the 64 possible nucleotide triplets and the edges join triplets that differ by one nucleotide under different arrangements of the nucleotides in a square (Figure 1). Each of the three possible arrangements of the nucleotides in the square yields different orderings of the codon triplets in the hypercube [33]; a fourth arrangement of the nucleotides is given by the square with its two diagonals, representing a scenario where all possible nucleotide changes are within reach in one step mutation. A more detailed description of the four possible arrangements and their corresponding modes of evolution has been reported elsewhere [31]. All 64 triplets can be represented in a 6D hypercube by considering the four possible arrangements in a square of the four nucleotides [7,15]. The SGC can be readily visualized as a graph of vertices, representing the codons, and edges, joining the codons at the Hamming distance of one. Thereby, the symmetries of the SGC can be obtained from the group of automorphisms of the graph [15].

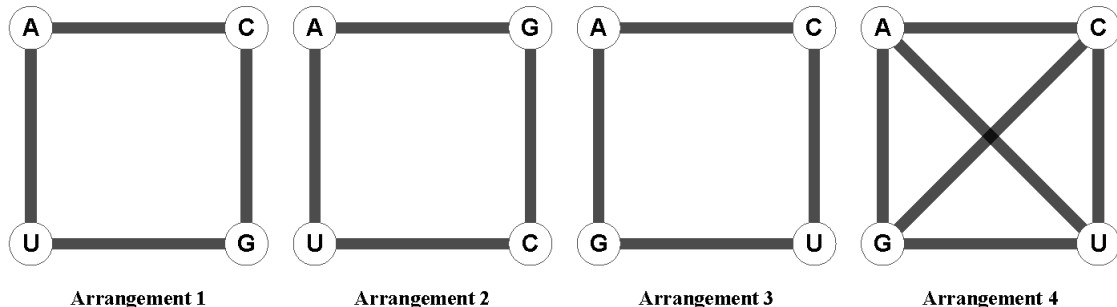

**Figure 1.** Four possible arrangements of the four nucleotides (A, C, G, U) as the vertices of a square. The four arrangements are not symmetrically equivalent.

The 6D model of the SGC is further transformed into its corresponding amino acid PHG through the algebraic quotient of the 6D model as a graph with the equivalence relation given by the assignation of codons to its corresponding amino acids [27,33,34]. The codons assigned to stop signals are removed from the hypercube model to correctly produce a PHG of the S-tRNA-C. In a phenotypic graph the vertices identify the set of amino acids, and any two given amino acids are joined by an edge if in the 6D hypercube codon model there exists codons codifying for such amino acids that were previously joined. The symmetries of a PHG are given by the group of graphs automorphisms which are all the bijective transformations of a graph to itself that preserves adjacencies.

The construction of the codon hypercube model and its corresponding PHG is also computed for the set of anticodon triplets which consist of the set of reverse complementary triplets of the codons with the removal of the anticodon triplets starting with adenine and the anticodons corresponding to the codons for the stop signal, thus resulting in a set of 45 anticodons.

The 6D codon model coupled with the PHGs has been used to calculate the probability transition matrix of a stochastic process that models amino acid substitutions given by a neutral model on which the nucleotide changes are at random [31]. We remark that the neutral evolution model and the neutrality test are as universal as the SGC [31]. The three stochastic processes using the three possible arrangements of the nucleotides in the square without diagonals are calculated and averaged. The stationary distribution of the averaged stochastic process determines the probability of finding an amino acid, provided that the protein mutates without selection pressures. The averaged stochastic

process approximates the stochastic process produced from the arrangement of the nucleotides in the square with its two diagonals [31].

Three synthetic genetic codes are designed (Figure 2) maintaining the principal properties of the SGC which are the codonicity of all the amino acids and the wobble property in the third nucleotide. The positioning of the stop codons in the synthetic codes was made by considering the stop signal as another signal encoded by the genetic code, i.e., as if it were another amino acid and thus had all their associated codons grouped together in the synthetic codes 1 and 3, while for the case of the synthetic code 2 we maintained the split in its codon block. The automorphism groups of their PHGs are computed for the four arrangements of the nucleotides. The stochastic process and stationary distributions of each code are further derived.

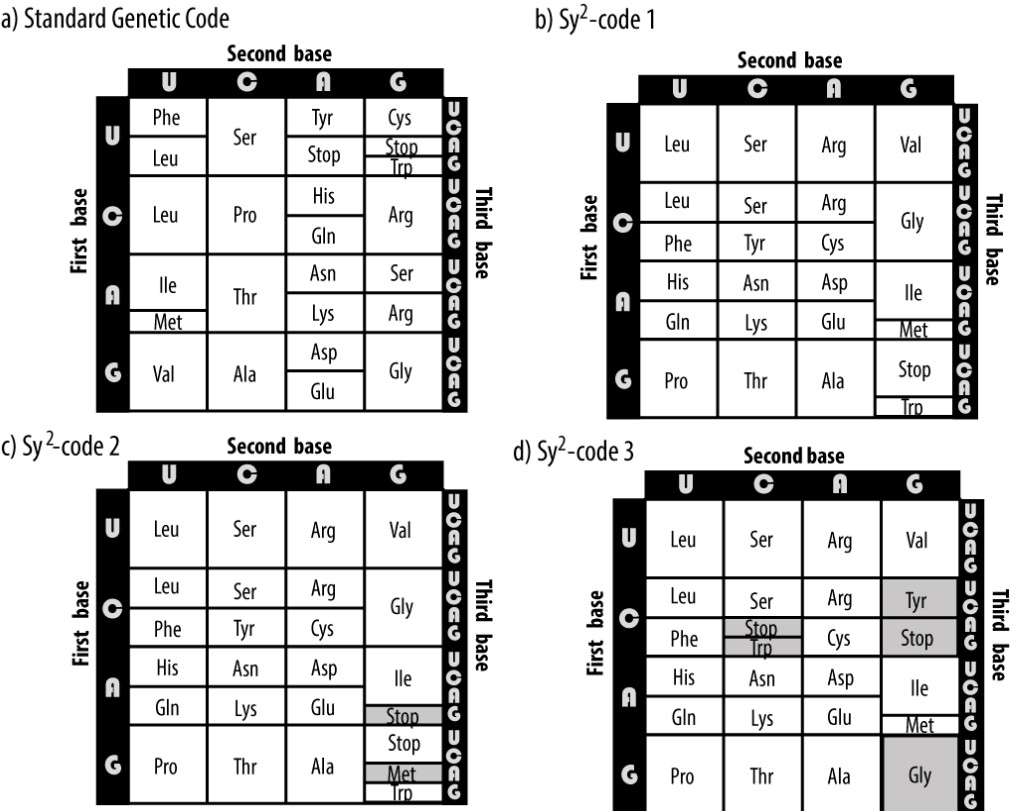

**Figure 2.** The standard genetic code (SGC) and three synthetic symmetrical genetic codes. The standard genetic code table (**a**) and the tables of three symmetric synthetic genetic codes. Changes from the synthetic code 1 (**b**) to the synthetic code 2 (**c**) and the synthetic code 3 (**d**) are shaded in grey.

## 3. Results

The PHGs for the SGC, and the synthetic codes using both codons and anticodons were computed using the four possible arrangements of nucleotides and analyzed for their symmetries (Table 1). The PHGs for the SGC for codons and anticodons for all the arrangements have as symmetry group the trivial group $e$, which means that there are no symmetric transformations other than the identity transformation. The PHGs for the synthetic code 1 on both codons and anticodons have as symmetry, the group $\mathbb{Z}_2$ for the arrangements of nucleotides without diagonals (arrangements 1, 2, and 3), whereas for the arrangement with diagonals (arrangement 4) the symmetry group is given by $S_3$. For the synthetic code 2 the symmetry groups of PHGs rendered by the codons and anticodons differ when considering the arrangement 2 of the nucleotides: the PHG of the codons has as symmetry the group $\mathbb{Z}_2^2$, whereas for the anticodon PHG the symmetry group is $\mathbb{Z}_2$. The symmetry groups coincide for the other nucleotide arrangements. For the synthetic code 3 the codon and anticodon PHGs coincide, with only the arrangement 2 without symmetries and the other nucleotide arrangements

having the symmetry group given by $\mathbb{Z}_2$. The group given by $\mathbb{Z}_2$ geometrically represents a reflection through an axis, the group $\mathbb{Z}_2^2$ represents the symmetries of a rectangle, whereas the group $S_3$ represents the permutations of a set with three elements or the symmetries of an equilateral triangle.

**Table 1.** Symmetry groups.

| | Standard Genetic Code | Synthetic Code 1 | Synthetic Code 2 | | Synthetic Code 3 |
|---|---|---|---|---|---|
| | Phenotypic graph of codons/anticodons | Phenotypic graph of codons/anticodons | Phenotypic graph of codons/anticodons | | Phenotypic graph of codons/anticodons |
| **Arrangement 1** | $e$ | $\mathbb{Z}_2$ | $\mathbb{Z}_2^2$ | | $\mathbb{Z}_2$ |
| **Arrangement 2** | $e$ | $\mathbb{Z}_2$ | $\mathbb{Z}_2^2$ | $\mathbb{Z}_2$ | $e$ |
| **Arrangement 3** | $e$ | $\mathbb{Z}_2$ | $\mathbb{Z}_2$ | | $\mathbb{Z}_2$ |
| **Arrangement 4** | $e$ | $S_3$ | $S_3$ | | $\mathbb{Z}_2$ |

The sets of orbits of each PHG under the action of its symmetry group are shown in Table 2. For the PHGs of codons and anticodons of the SGC, there are no orbits in any of the four arrangements. For the synthetic symmetrical codes, note that the codonicity of the amino acids grouped through the action of the symmetry group is the same, i.e., given an amino acid, its orbit under the action of the group contains amino acids which have the same codonicity. The latter does not hold true for the synthetic code 3, where the graphs of codons and anticodons are asymmetric under arrangement 2.

**Table 2.** Sets of orbits.

| | Standard Genetic Code | Synthetic Code 1 | Synthetic Code 2 | Synthetic Code 3 |
|---|---|---|---|---|
| | Phenotypic graph of codons/anticodons | Phenotypic graph of codons/anticodons | Phenotypic graph of codons/anticodons | Phenotypic graph of codons/anticodons |
| **Arrangement 1** | N/A | {N, H}, {Q, K}, {L, S}, {F, Y}, {P, T} | {N, H}, {Q, K}, {L, S}, {F, Y}, {P, T} | {N, H}, {Q, K}, {L, S}, {F, W}, {P, T} |
| **Arrangement 2** | N/A | {A, T}, {R, S}, {N, D}, {C, Y}, {E, K} | {A, T}, {R, S}, {N, D, E, K}/{A, T}, {R, S}, {N, D}, {C, Y}, {Q, H}, {C, Y}, {E, K} | N/A |
| **Arrangement 3** | N/A | {A, P}, {R, L}, {D, H}, {C, F}, {Q, E} | {A, P}, {R, L}, {D, H}, {C, F}, {Q, E} | {A, P}, {R, L}, {D, H}, {C, F}, {Q, E} |
| **Arrangement 4** | N/A | {A, P, T}, {R, L, S}, {N, D, H}, {C, F, Y}, {Q, E, K} | {A, P, T}, {R, L, S}, {N, D, H}, {C, F, Y}, {Q, E, K} | {A, P}, {R, L}, {D, H}, {C, F}, {Q, E} |

Mono-codonic: W (Trp), M (Met); di-codonic: Y (Tyr), C (Cys), E (Glu), K (Lys), Q (Gln), H (His), F (Phe), N (Asn), D (Asp); tri-codonic: I (Ile); tetra-codonic: A (Ala), T (Thr), G (Gly), P (Pro), V (Val); and hexa-codonic: L (Leu), S (Ser), and R (Arg).

The codon and anticodon graphs facilitate the analyses of the evolvability of the genetic code, including tailored-design codes. The stationary distributions of the stochastic processes retrieved by each genetic code are shown in Figure 3. In the stationary distribution for the synthetic code 1, the probability of the amino acids with the same codonicity is approximately the same with a variance of $7.29 \times 10^{-8}$ for the amino acids with two codons (Tyr (Y), Cys (C), Glu (E), Lys (K), Gln (Q), His (H), Phe (F), Asn (N), Asp (D)); $2.12 \times 10^{-10}$ for the amino acids with four codons (Gly (G), Pro (P), Thr (T), Val (V), Ala (A)); and $2.76 \times 10^{-7}$ for the amino acids with six codons (Leu (L), Ser (S), Arg (R)). This pattern is not present for the synthetic codes 2 and 3. For the synthetic code 2 the amino acids as Glu, Lys, Gln, Gly, and Val have lower probabilities than the other amino acids with the same codonicities. For the synthetic code 3 the amino acids Tyr, Cys, Phe, Gly, Val, and Ser have lower probabilities than the other amino acids with the same codonicities. The equiprobable behavior of the stationary distribution of the synthetic code 1 is not present in the stationary distribution of the SGC where the probabilities of Tyr, Cys, Glu, Lys, Gln are lower than the probabilities of His, Phe, Asn, Asp for the di-codonic amino acids. On the amino acids with four codons the probabilities of Gly and Pro are lower than the probabilities of Thr, Val, Ala. For the hexa-codonic amino acids the probability of Arg is greater than the probability of Leu and Ser. On the synthetic codes and the SGC the probabilities for the uni-coded amino acids Trp and Met are significantly different due to the removal of the codons

for the stop signal in the construction of the PHGs. Exact values of the stationary distributions are provided in Supplementary Information.

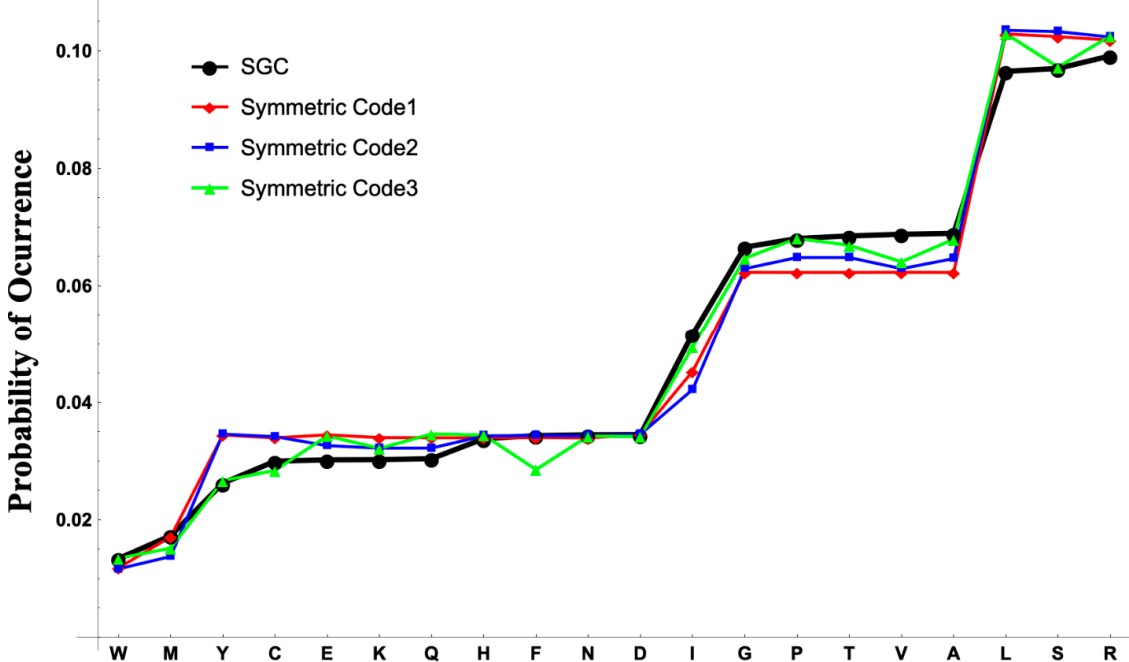

**Figure 3.** The Amino Acid Neutrality Test Applied to a Hypothetical Protein Obeying the SGC or the Synthetic Codes. The amino acids are ordered according to their codonicity, i.e., from mono-codonic: W (Trp), M (Met); di-codonic: Y (Tyr), C (Cys), E (Glu), K (Lys), Q (Gln), H (His), F (Phe), N (Asn), D (Asp); tri-codonic: I (Ile); tetra-codonic: G (Gly), P (Pro), T (Thr), V (Val); A (Ala); and hexa-codonic: L (Leu), S (Ser), and R (Arg).

## 4. Discussion

In this work we showed that the PHGs of codons and anticodons as obtained from the SGC are asymmetric for all possible arrangements of the nucleotides. This result is in stark contrast with the symmetries found in the SGC [5–8]. To elucidate the meaning of the asymmetry observed, three synthetic and symmetrical codes with non-trivial symmetries on the mathematical representation of codons and anticodons were designed. We found that the corresponding PHGs of codons and anticodons exhibited symmetry except for arrangement 2. We also observed that in the built-in symmetric codes the amino acids were grouped in similar orbits according to their codonicity, i.e., the amino acids contained in the same orbit have the same codonicity. In contrast, in the natural PHGs the amino acids are orbit-free. Recall that the degeneracy of both the SGC and the synthetic symmetrical codes is the same. Yet, the PHGs of the SGC are asymmetric whereas most PHG of the synthetic symmetrical codes are symmetrical. Next, we subjected the four codes to a neutral model of protein evolution and calculated their stationary distributions. The probability of occurrence of the amino acids for the synthetic codes have a constant probability determined by the redundancy of the amino acid. For the case of the synthetic code 1, whose symmetries are non-trivial for all the nucleotide arrangements and the same codon and anticodon representation in 6D, the amino acids on the same orbit have the same probability of appearance as shown by their respective stationary distribution of the amino acid substitution process. This phenomenon is not as clear for the synthetic code 3, where the symmetry group for the codon and anticodon representation is the trivial group. The neutral test is the null hypothesis of evolution, and it clearly discerns putative amino acids with unexpected frequencies that might be under positive, negative selection, or neutral. Therefore, the $\frac{d_N}{d_S}$ ratio, i.e., the number of nonsynonymous substitutions to the number of synonymous substitutions will

be different for the different codes. In symmetric graphs, the amino acid changing (non-synonymous) and amino acid conserving (synonymous) nucleotide sites might evolve at similar rates if they pertain to the same orbit.

In asymmetric graphs, the choice of an amino acid is not enslaved to the choice of any other amino acid. The asymmetry observed is an ancient property that left intact the universality of the SGC and the selective forces that shaped the evolving codes. The asymmetry was frozen since at least the Extended RNA codes [7]. The SGC terminates its influence after the aminoacylation of each tRNA. Asymmetry of PHGs means that amino acids have more degrees of freedom for exploring the sequence space for innovation in protein evolution. Afterwards, the asymmetry of PHGs of codons and anticodons is no longer influenced by the SGC. Asymmetry of the PHGs of codons and anticodons have facilitated the astonishing diversity of living organisms. We contend that several symmetrical codes were formed during evolution, but only the one(s) that had asymmetrical PHGs of codons and anticodons prevailed. The lack of symmetry in the PHGs of the SGC as compared to the ones of the synthetic codes shows that the usual SGC table is visually well-organized by grouping the amino acids in boxes, where the distribution of such boxes is not symmetric. In addition, the organization of amino acids in boxes is given by the wobble effect and the degeneration of the SGC. The result of the stationary distribution from the SGC and the synthetic codes shows that if the distribution of the amino acids in the SGC, i.e., the boxes of the amino acids was symmetric, the frequency of occurrence of different amino acids would not be independent of each other given neutral point mutations. In fact, the implementation of the genetic code would be biased to maintain similar frequencies of the amino acids with the same codonicity. This relation of symmetry/asymmetry describes two features of the genetic code. The symmetric properties of the SGC allow for robustness, for it to be less error-prone in its structure, whereas the asymmetric feature grants independence to the amino acids in protein evolution. The codonicity of an amino acid affects its codon usage bias which has been shown to be in co-evolution with the tRNA gene composition and it is in agreement with the selection-mutation-drift theory of codon usage in the optimization of translation [35].

In comparison with the SGC, the symmetry of the PHGs is broken by the absence of 5'A anticodons and of the anticodons that correspond to the stop codons. Note that sy2-codes have the same degeneracy of the SGC and yet some of them displayed symmetrical graphs. Therefore, degeneracy and asymmetry work together to achieve the optimality of the SGC [36]. The PHGs of codons and anticodons of mitochondrial codes turned out to be also asymmetric (not shown) [33]. Symmetry allows regularities but asymmetries allow evolvability. SGC is the result of an orchestrated coevolution of several molecules as mentioned in introduction. Life possess the salient feature of being formed by a plethora of codes that operate at different scales.

A review of group theory and abstract algebra applied to molecular systems biology can be found in [37]. These theoretical approaches can enlighten complex problems in biology. Mathematics in biology has often been regarded as an intruder perhaps for its descriptive character of the latter. Yet, it is necessary to take cognizance of the fact that most biological signals and biological systems are complex. Mathematics has become pervasive in all areas in biology. Collaboration through interdisciplinary analyses can provide new insights to complex problems such as the origin and evolution of the SGC and proteins.

**Supplementary Materials:** The Supplementary Materials are available online at http://www.mdpi.com/2073-8994/12/6/997/s1.

**Author Contributions:** M.V.J. and G.S.Z. conceived the whole work, contributed with ideas; M.V.J. and G.S.Z. performed the analyses; M.V.J. coordinated the research. M.V.J. and G.S.Z. wrote the manuscript. All authors have read and agreed to the published version of the manuscript.

**Funding:** M.V.J. was funded by Dirección General de Asuntos del Personal Académico (DGAPA), Universidad Nacional Autónoma de México, UNAM (PAPIIT-IN201019); G.S.Z. is a doctoral student from Programa de Doctorado en Ciencias Biomédicas, Universidad Nacional Autónoma de México (UNAM) and received a doctoral fellowship from CONACYT (number: 737920).

**Acknowledgments:** We thank Francisco Prosdocimi for his critical comments and Juan R. Bobadilla for material support.

**Conflicts of Interest:** The authors declare no conflict of interest.

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
