# Peer review of "On the Importance of Asymmetry in the Phenotypic Expression of the Genetic Code upon the Molecular Evolution of Proteins"

_symmetry, doi:10.3390/sym12060997_

Round 1

Reviewer 1 Report

In the present study, the authors revealed how the asymmetry, which is defined in the manuscript, of the standard genetic code influences protein evolution in the neutral evolution model. Their study is purely theoretical, but the authors are careful in preserving the important properties of the natural code in their calculations: the codon degeneracy, or “codonicity” in their term, and the number of tRNA anticodons. Thus, the synthetic codes in Figure 2 all have the same number of codons for each amino acid, and could be translated by the same number of tRNAs, as in the natural code. In this regard, the authors tried to make their synthetic codes look realistic.

My concern is that the authors do not seem to be as careful about the molecular basis for how the three stop codons are recognized by cellular factors and signal translational termination.

The real stop codons are UAA, UAG, and UGA, sharing U in the first position and purine in the second. In bacteria, two protein factors (release factors 1 and 2) exist to recognize these codons: one specific for UAA and UAG, and another for UAA and UGA. The eukaryotic release factors recognize all of the three stop codons. Thus, the release factors recognize U and purine in the first and second, respectively, positions of the codons. Correspondingly, stop codons should be conservative about the first position and semi-conservative about the second position, and only one of the artificially constructed codes (Figure 2) observes this rule. I feel that the authors need to make any comments on how they treated stop codons when they constructed the model codes.

A minor point:

“Codonicity” should be defined in the Abstract, or rephrased to common words.

Author Response

REVIEWER 1

In the present study, the authors revealed how the asymmetry, which is defined in the manuscript, of the standard genetic code influences protein evolution in the neutral evolution model. Their study is purely theoretical, but the authors are careful in preserving the important properties of the natural code in their calculations: the codon degeneracy, or “codonicity” in their term, and the number of tRNA anticodons. Thus, the synthetic codes in Figure 2 all have the same number of codons for each amino acid and could be translated by the same number of tRNAs, as in the natural code. In this regard, the authors tried to make their synthetic codes look realistic.

 My concern is that the authors do not seem to be as careful about the molecular basis for how the three stop codons are recognized by cellular factors and signal translational termination.

The real stop codons are UAA, UAG, and UGA, sharing U in the first position and purine in the second. In bacteria, two protein factors (release factors 1 and 2) exist to recognize these codons: one specific for UAA and UAG, and another for UAA and UGA. The eukaryotic release factors recognize all of the three stop codons. Thus, the release factors recognize U and purine in the first and second, respectively, positions of the codons. Correspondingly, stop codons should be conservative about the first position and semi-conservative about the second position, and only one of the artificially constructed codes (Figure 2) observes this rule. I feel that the authors need to make any comments on how they treated stop codons when they constructed the model codes.

We do agree with the reviewer that we left undefined the type(s) of release factors that could recognize the stop codons in the synthetic codes. We did so because this issue does not impinge in the phenotypic graphs of anticodons or amino acids. Yet, we remarked in lines 135-138 of Material and Methods the following:

“The positioning of the stop codons in the synthetic codes was made by considering the stop signal as another signal encoded by the genetic code, i.e., as if it were another amino acid and thus had all their associated codons grouped together in the synthetic code 1 and 3, while for the case of the synthetic code 2 we maintained the split in its codon block.”

A minor point:

 “Codonicity” should be defined in the Abstract, or rephrased to common words.

We modified the following sentence in the Abstract:

“In the symmetrical synthetic codes, the amino acids are grouped according to their codonicity, this is, the number of triplets/codons encoding a given amino acid.”

Submission Date

28 April 2020

Date of this review

30 Apr 2020 08:59:10

Reviewer 2 Report

In this manuscript, authors detected the asymmetry of phenotyping graphs of codons and anticodons from the Standard Genetic Code (SGC). They discussed the possible biological meaning of such an asymmetry.

Overall, the MS is well written and could be of interest to the specialist studying the symmetry phenomena.

Although, I would suggest expending the Discussion part about the meaning of SGC degeneracy and asymmetry in protein evolution e.g. in relation to codon bias in different organisms.

It would be good to give a reference to where the term “codonicity” (line 22) was first introduced (ref. on like 77) since it is not a widely used term. Also, the readability of the MS would increase if authors would briefly introduce other basic terms for this work e.g. phenotypic graphs, orbits, etc.

Lines 97 and 134. Figure 1 is identical to Figure 1 from (Zamudio et al, 2019) ref. [31]. I suggest giving a proper reference here, especially because in (Zamudio et al, 2019) the detailed explanation was given already.

Author Response

In line 264 of Discussion we added the following:

“The codonicity of an amino acid affects its codon usage bias which has been shown to be in co-evolution with the tRNA gene composition and it is in agreement with the selection-mutation-drift theory of codon usage in the optimization of translation [35].”

We added the following reference:

  1. Rocha, E.P.C. Codon Usage Bias from tRNA’s point of view: Redundancy, specialization, and efficient decoding for translation optimization. Genome research. 2004,14, 2279-86.

We thank the reviewer for her/his idea.

It would be good to give a reference to where the term “codonicity” (line 22) was first introduced (ref. on like 77) since it is not a widely used term. Also, the readability of the MS would increase if authors would briefly introduce other basic terms for this work e.g. phenotypic graphs, orbits, etc.

Regarding the term “codonicity”, we modified the following sentence in the Abstract:

“In the symmetrical synthetic codes, the amino acids are grouped according to their codonicity, this is, the number of triplets/codons encoding a given amino acid.”

In lines 112-114 we provide a definition of what a phenotypic graph is:

“The 6D model of the SGC is further transformed into its corresponding amino acid phenotypic graph (PHG) through the algebraic quotient of the 6D model as a graph with the equivalence relation given by the assignation of codons to its corresponding amino acids [27,33,34].”

We added the formal definition of an orbit in Lines 88- 92:

Remark: When a group  is operating over a set  as a group of transformations of  we call orbit of an element  of  the set of all the elements that are images of  under the action of  The set of orbits is a partition of E associated to the equivalence relations determined by the action of  Two elements are equivalent under this action if each is the image of the other under a transformation determined by

Lines 97 and 134. Figure 1 is identical to Figure 1 from (Zamudio et al, 2019) ref. [31]. I suggest giving a proper reference here, especially because in (Zamudio et al, 2019) the detailed explanation was given already.

In lines 106-107, we added:

“A more detailed description of the four possible arrangements and their corresponding modes of evolution has been reported elsewhere [31].”

Submission Date

28 April 2020

Date of this review

12 May 2020 21:27:27

We are grateful to the reviewers for their helpful comments and criticisms. They have helped us to improve the quality and presentation of the paper. We expect that the present version of the manuscript answers all their concerns.

Reviewer 3 Report

This is a new and interesting article proposed by José and Zamudio about the genetic code visualized from a mathematical point of view. This work is in the continuation of a number of previous publications by the same authors on the same topic. Here, they calculate the importance of the internal structure of codes on their robustness against mutations (not a new problem) and the independence of amino acids during evolution. To do this, they have compared the standard code (asymmetrical) with three theoretical symmetrical codes. Their results looks straightforward. My only concern is that the article, like the previous ones of the same authors, is unlikely to be read and understood by many biologists, given its mathematical language. They mention that in their introduction. The authors are talking about hypercube, group theory, orbits, graph, vertices, etc .... Albeit classical in mathematics, I am afraid that these notions are foreign to biologists. If possible, I think it would be an interesting idea to try to make them more explicit if the authors want that their work is taken into consideration by the biologists. 
